# Determining the Effects of Light on the Fruit Peel Quality of Photosensitive and Nonphotosensitive Eggplant

**DOI:** 10.3390/plants11162095

**Published:** 2022-08-12

**Authors:** Zhaoze Sang, Jinhua Zuo, Qing Wang, Anzhen Fu, Yanyan Zheng, Yonghong Ge, Zongwei Qian, Yanling Cui

**Affiliations:** 1Key Laboratory of Vegetable Postharvest Processing, Ministry of Agriculture and Rural Affairs, Beijing Key Laboratory of Fruits and Vegetable Storage and Processing, Key Laboratory of Biology and Genetic Improvement of Horticultural Crops (North China) of Ministry of Agriculture, Key Laboratory of Urban Agriculture (North) of Ministry of Agriculture, Institute of Agri-Food Processing and Nutrition, Beijing Vegetable Research Center, Beijing Academy of Agriculture and Forestry Sciences, Beijing 100097, China; 2College of Food Science and Technology, Bohai University, Jinzhou 121013, China; 3National & Local Joint Engineering Research Center of Storage, Processing and Safety Control Technology for Fresh Agricultural and Aquatic Products, Bohai University, Jinzhou 121013, China

**Keywords:** eggplant, photosensitive, nonphotosensitive, light, transcriptome

## Abstract

With the development of facility agriculture, low-light stress is a prominent problem and a popular research topic currently. In this study, transcriptome analysis was used to analyze the genes in the fruit peel of photosensitive and nonphotosensitive eggplant and to explore the mechanism of changes in fruit color, texture, hormone content, aroma, and taste of these two different types of eggplant. We identified 51, 65, 66, and 66 genes involved in synthesizing anthocyanins, texture, hormone content, and aroma and flavor, respectively, in the two different types of eggplant based on the variation in gene expression trends in the fruit peel. These results provide a basis for further analysis of the molecular mechanism underlying the regulatory processes in eggplant fruits under low-light stress.

## 1. Introduction

Eggplant (*Solanum melongena* L.) is a common vegetable species that has been grown in China for nearly 1000 years and is the fourth most widely planted Solanaceae vegetable species after potato, tomato, and pepper [1,2]. Eggplant is considered to be one of the ten most important vegetable crops in the world [3,4,5], with considerable importance in Asia and the Mediterranean region [6,7]. Eggplant is a high-yielding cash crop species with extensive prospects for development and utilization, and its fruits have rich nutritional value [8]. Eggplant fruit is rich in minerals, protein, antioxidants, etc., and is often used in daily meals [9,10,11]. The color of eggplant fruit is a vital commodity trait; only when an eggplant fruit displays its proper color can it be competitive on the market. The main colors of eggplant fruits on the market are purple-black and fuchsia, which are very popular.

Fruit ripening is a complex process influenced by internal and external factors to varying degrees, leading to a series of metabolic processes that ultimately determine fruit color, texture, flavor, and aroma [12]. Light is a basic and essential factor that causes physiological changes in plants and plays an important role in the whole process of plant growth and development [13]. Therefore, autotrophic plants avoid darkness and undergo phototropism [14]. In addition, there were significant differences in fruit texture-related genes between light and dark environments [15]. Schuster et al. [16] found that cellulase transcription levels increased over time in dark environments but only briefly under light conditions. Fruit softening can lead to a profound loss of quality during storage [17]. Pombo et al. [18] found that illumination played a promoting role in maintaining the firmness of strawberry fruits. Light also has a certain influence on the content of plant hormones. Jiang et al. [19] found that low light treatment reduced the contents of salicylic acid, jasmonic acid, and zeatin in tomatoes. Low light also has a certain effect on fruit and vegetable flavor. Sucrose and its degradation enzyme activities in rice grains decreased after low light treatment [20]. In general, a low light environment has many effects on plants, but for eggplant, peel color change is the most significant [2].

Eggplant is rich in anthocyanins, which are natural plant pigments that determine the color of flowers and fruits [21]. The main anthocyanin in eggplant is delphinidin-3-(p-coumaroyl rutinoside)-5-glucoside (nasunin) [22]. Sadilova et al. [23] found that delphinin-3-rupanin, which is abundant in eggplant, is highly stable, can scour free radicals well, and can resist oxidation. The biosynthesis of anthocyanins is one of the branches of the metabolic pathway of phenylalanine [24]. Phenylalanine is catalyzed by a series of enzymes to produce unstable anthocyanins, which are then stabilized by acylation (AT), methylation (MT), glycosylation (GT), and hydroxylation (HT) [25,26]. The synthesis of anthocyanins in plants is affected by many factors, such as sugar content, light, and temperature, of which light is more important [27,28,29]; light can affect anthocyanin synthesis in plants by regulating the expression of the R2R3-MYB gene [30,31].

During the long-term evolution of plants, a series of complex regulatory networks formed to sense environmental signals [32,33]. The anthocyanin biosynthesis network consists of upstream regulatory genes and downstream regulatory genes [34]. Currently, most of the studies on anthocyanin biosynthesis genes focus on the MBW complex composed of *MYB* transcription factors, *bHLH* transcription factors, and *WD40* proteins. After sensing the light signal, plant photoreceptors such as phytochrome, *PHY*, *PHOT*, *CRY*, and *UVR8* transmit the signal to downstream transcription factors to promote or inhibit the expression of structural genes [35]. *Constitutive photomorphogenesis protein 1* (*COP1*) is a photomorphogenetic factor [36]. In Arabidopsis, *COP1* is also considered a “central regulator” of light signal transduction due to its interaction with both upstream photoreceptor proteins and downstream target proteins [37]. Under dark conditions, *COP1* is present in the nucleus [38]. *COP1* promotes photomorphogenesis and the degradation of transcription factors. Under light conditions, the concentration of *COP1* in the nucleus decreases rapidly [39], and photoactivated photoreceptors inhibit the activity of *COP1*, leading to photomorphogenesis, which in turn promotes the accumulation of transcription factors. Maier et al. [40] reported that the *COP1/SPA* complex controls the anthocyanin levels in Arabidopsis, enabling anthocyanin synthesis even in the dark environment.

As an essential cash crop product, eggplant fruit is greatly enjoyed by humans [41]. Eggplant fruit has high medicinal and health value, and eggplant yields are increasing annually worldwide [42]. However, low-temperature and low-light conditions often lead to the poor coloration of eggplant fruit [35,43]. In actual production, it was found that under dark environments, some eggplants synthesized anthocyanins in their peel [2]. Honda et al. [44] found through hybridization experiments that the genes controlling the photosensitive types and the main types of anthocyanins in eggplant are independent of each other and that the genes controlling the photosensitive eggplant types are dominant. In this experiment, photosensitive and nonphotosensitive eggplant were used as research materials to explore the similarities and differences between these eggplant types in terms of their resilience to shading treatment. After bagging the eggplant fruits, we performed an RNA sequencing analysis to identify the genes related to color, flavor, texture, and hormone changes in the eggplant fruit peel after dark conditions were applied. We compared the transcriptome data for the peels of photosensitive eggplant and nonphotosensitive eggplant at the same time point to explore the effects of darkness and illumination on the color-, texture-, flavor-, and hormone synthesis-related genes in the fruits. We aimed to identify some key genes that can reduce the effects of low-light environments, improve the quality of eggplant fruits, and provide new ideas and methods for the cultivation and breeding of eggplant.

## 2. Materials and Methods

### 2.1. Plant Materials and Treatment

The late-generation eggplant inbred lines 20Q109 and 20Q111 were used for transcriptional analysis and population construction. Line 20Q109 is photosensitive; its fruits are purple-black, with a purple-black peel and a long stick-like shape. After bagging, the fruit peel of this inbred line is pure white, but after reillumination, the peel color returns to purple-black. Line 20Q111 is nonphotosensitive; its fruits are purple-black, with a purple-black peel and a long stick-like shape. The fruits of this line are light purple-red after bagging but become purple-black after reillumination.

We set up three treatment groups: CK1, T1, and T2 (photosensitive eggplant is represented by P and nonphotosensitive eggplant by NP). Group CK1 represents eggplant fruits without any treatment of natural light. Group T1 was immediately bagged for shading after pollination, treated with shading for 20 days. Group T2 represents that after shading eggplant for 20 days, it is illuminated for 5 days (shading bag: A yellow paper bag with a black inner layer that is opaque to light).

Eggplant skin samples (length 3 cm, width 1 cm, thickness 0.1–0.15 cm) are collected with peeler and frozen with liquid nitrogen. The frozen eggplant peel samples were used in anthocyanin and transcriptome experiments, and each treatment group had three biological repeats.

### 2.2. Determination of Anthocyanin Content

The peel of eggplant was shredded, frozen with liquid nitrogen, ground into powder with a grinding machine, and weighed 5 g sample powder into a 100 mL beaker. then, 20 mL anhydrous ethanol, 5 mL 0.1 mol/L citric acid and 15 mL distilled water was added into the beaker successively; it underwent ultrasonic treatment for 80 min; was centrifuged at 10,000 r/min speed for 10 min; after centrifugation, the supernatant was taken to measure the UV absorption of the diluted solution at 530 nm, and the anthocyanin content in the sample to be measured was calculated.

### 2.3. RNA Extraction and Library Construction

An RNA extraction kit was used to extract RNA samples (rn40, Aidlab Biotechnology, Beijing, China). RNA concentration and purity were then measured using a Nanodrop 2000 instrument (Thermo Fisher Scientific, Wilmington, DE, USA). To ensure that the samples to be sequenced were of sufficient quality, RNA integrity was assessed using an Agilent 2100 Bioanalyzer (Agilent Technologies, Santa Clara, CA, USA) and an RNA Nano6000 detection kit. After the RNA samples were qualified, libraries were constructed. First, eukaryotic mRNA was enriched with oligo (DT) magnetic beads. Then, a fragmentation buffer was added to randomly interrupt the mRNA. Using mRNA as a template, first-strand cDNA was synthesized with random hexamers. Then, second-strand cDNA was synthesized by adding buffer, dNTPs, RNase H, and DNA polymerase I. The cDNA was subsequently purified by Ampure XP beads. The purified double-stranded cDNA was subjected to end repair, and an A tail was added and connected to a sequencing adaptor, after which the fragment size was selected via Ampure XP beads. Finally, to ensure library quality, qPCR was performed to accurately determine the effective concentration of the library (the effective concentration of the library components >2 nm) [45,46].

### 2.4. Sequence Assembly and Gene Annotation

Low-quality reads, adaptor-contaminated reads, and reads with a high content of unknown bases (N) were removed via quality control (QC) before downstream analyses. After read filtering, the clean reads were mapped to the reference genome by HISAT. The eggplant reference genome (consortium V3) database (https://solgenomics.net/organism/Solanum_melongena/genome (accessed on 27 May 2022)) was used as a reference [47]. After the novel transcripts were obtained, their coding transcripts were merged with reference transcripts to create the complete reference sequence, which was used for gene expression analysis.

### 2.5. Identification and Annotation of Differentially Expressed Genes (DEGs)

After the establishment of the complete reference sequence, the clean reads were mapped to the reference library using Bowtie2; then, the gene expression level for each sample was calculated by RSEM. DEGs were detected by DESeq2, and a fold-change ≥2.00 and an adjusted *p* value ≤ 0.05 were used as criteria. Gene Ontology (GO) classification and functional enrichment were performed by WEGO software (https://biodb.swu.edu.cn/cgi-bin/wego/index.pl (accessed on 27 May 2022)). Pathway classification and functional enrichment of the DEGs were performed via the Kyoto Encyclopedia of Genes and Genomes (KEGG) database [48]. Cluster analysis of expression patterns was performed by MeV (4.9).

### 2.6. Statistical Analysis of the Data

Excel 2019 software was used for basic data sorting. Analysis of variance was conducted using Statistical Analysis System (SAS 9.3) (SAS Institute, Inc., Cary, NC, USA). Each group of gene expression data contained three replicates, and the average value was taken to calculate the standard error. The data were plotted using Origin 7.0 software (Origin Lab, Northampton, MA, USA).

## 3. Results

### 3.1. Anthocyanin Content in Photosensitive Eggplant and Nonphotosensitive Eggplant

After measurement, it was found that there was a big difference in anthocyanin content between photosensitive eggplant and nonphotosensitive eggplant peel (Table 1). We found that the average content of anthocyanin in photosensitive eggplant peel before shading treatment was 20.4, while the average content of anthocyanin in nonphotosensitive eggplant peel was 64.6, indicating that the anthocyanin content in nonphotosensitive eggplant peel was 216.66% higher than that in photosensitive eggplant peel.

### 3.2. Screening of Differentially Expressed Genes (DEGs) between Photosensitive Eggplant and Nonphotosensitive Eggplant

After shading and light treatment, the appearance of the two kinds of eggplant was significantly changed (Figure 1A,B); thus, the differential expression of genes in eggplant peel were analyzed. We ultimately identified a total of 1733 DEGs (622 upregulated and 1111 downregulated) in the CK1-P vs. T1-P comparison group. A total of 3480 DEGs (1636 upregulated and 1844 downregulated) were found in the T1-P vs. T2-P comparison group, with 1626 DEGs (424 upregulated and 1202 downregulated) found in the CK1-NP vs. T1-NP comparison group, 2962 DEGs (1342 upregulated and 1620 downregulated) found in the T1-NP vs. T2-NP comparison group, 3480 DEGs (1636 upregulated and 1844 downregulated) found in the T1-P vs. T1-NP comparison group, and 2962 DEGs (1342 upregulated and 1620 downregulated) found in the T2-P vs. T2-NP comparison group (Figure 1C). By comparing the DEGs in each comparison group, we found that the gene expression of the photosensitive eggplant was different from that of the nonphotosensitive eggplant. Nevertheless, the gene changes in the two kinds of eggplant after shading treatment and reillumination were significant. On the whole, shading treatment had a greater effect on the photosensitive eggplant, as the gene changes in those samples were more apparent, and the differences in gene expression changes led to the differences in physiology between the photosensitive eggplant and nonphotosensitive eggplant.

### 3.3. Gene Ontology (GO) Analysis of DEGs

GO enrichment analysis was performed on the DEGs in the CK1-P vs. T1-P, T1-P vs. T2-P, CK1-NP vs. T1-NP, T1-NP vs. T2-NP, T1-P vs. T1-NP, and T2-P vs. T2-NP comparison groups. The GO annotation terms were divided into three categories: biological processes, cellular components, and molecular functions. DEGs related to biological processes accounted for the majority, and the top 20 GO annotations related to biological processes in each comparison group are shown in Figure 2A–F. In the CK1-P vs. T1-P comparison group, the significantly enriched GO annotations mainly included GO: 0009536 (plastid), GO: 0044435 (plastid part), and GO: 0009526 (plastid envelope) (Figure 2A). In the T1-P vs. T2-P comparison group, the significantly enriched GO annotations mainly included GO: 0009536 (plastid), GO: 0044435 (plastid part), and GO: 0009526 (plastid envelope) (Figure 2B). In the CK1-NP vs. T1-NP comparison group, the significantly enriched GO annotations mainly included GO: 0044435 (plastid part), GO: 0009526 (plastid envelope), and GO: 0009507 (chloroplast) (Figure 2C). In the T1-NP vs. T2-NP comparison group, the significantly enriched GO annotations mainly included GO: 0071944 (cell periphery), GO: 0030312 (external encapsulating structure), and GO: 0005576 (extracellular region) (Figure 2D). In the T1-P vs. T1-NP comparison group, the significantly enriched GO annotations mainly included GO: 0071944 (cell periphery), GO: 0030312 (external encapsulating structure), and GO: 0005576 (extracellular region) (Figure 2E). In the T2-P vs. T2-NP comparison group, the significantly enriched GO annotations mainly included GO: 0044435 (plastid part), GO: 0009507 (chloroplast), and GO: 0044434 (chloroplast part) (Figure 2F). GO analysis showed that DEGs were mainly enriched in cellular components in the six comparison groups. In addition, as shown in Figure 2, in photosensitive and nonphotosensitive eggplant, most of these differentially expressed genes were downregulated after shading treatment and upregulated after restoration of light.

### 3.4. Kyoto Encyclopedia of Genes and Genomes (KEGG) Pathway Enrichment Analysis of DEGs

To further understand the functions of the DEGs and determine the main pathways involved in the process of light–dark regulation in the fruits of eggplant with different levels of photosensitivity, KEGG enrichment analysis was performed for all differentially expressed genes (Figure 3). The results showed that in the CK1-P vs. T1-P comparison group, the genes involved in postharvest ripening of eggplant fruits were significantly enriched in metabolic pathways, the biosynthesis of secondary metabolites, carbon metabolism, and photosynthesis (Figure 3A). In the T1-P vs. T2-P comparison group, genes involved in postharvest ripening of eggplant fruits were significantly enriched in metabolic pathways, the biosynthesis of secondary metabolites, carbon metabolism, and hormone signal transduction (Figure 3B). In the CK1-NP vs. T1-NP comparison group, genes involved in postharvest ripening of eggplant fruits were significantly enriched in metabolic pathways, the biosynthesis of secondary metabolites, carbon metabolism, and phenylpropanoid biosynthesis (Figure 3C). In the T1-NP vs. T2-NP comparison group, genes involved in postharvest ripening of eggplant fruits were significantly enriched in metabolic pathways, the biosynthesis of secondary metabolites, carbon metabolism, and phenylpropanoid biosynthesis (Figure 3D). In the T1-P vs. T1-NP comparison group, genes involved in postharvest ripening of eggplant fruits were significantly enriched in metabolic pathways, the biosynthesis of secondary metabolites, protein processing in endoplasmic reticulum, and phenylpropanoid biosynthesis (Figure 3E). In the T2-P vs. T2-NP comparison group, genes involved in postharvest ripening of eggplant fruits were significantly enriched in metabolic pathways, the biosynthesis of secondary metabolites, carbon metabolism, and photosynthesis (Figure 3F). In conclusion, gene enrichment in metabolic pathways, the biosynthesis of secondary metabolites, and carbon metabolism was most significant in the six comparison groups.

### 3.5. Screening and Functional Identification of Different Genes in Each Group

The DEGs in each comparison group represented a series of expression changes in the genes in eggplant fruits under dark and light treatment, and the differential expression of these genes was a manifestation of the effects of light and dark conditions on eggplant fruits. Therefore, we focused on the expression patterns of these DEGs. The results showed that there were 372 (CK1-P vs. T1-P: 87 upregulated and 285 downregulated), 658 (T1-P vs. T2-P: 415 upregulated and 243 downregulated), 421 (CK1-NP vs. T1-NP: 45 upregulated and 376 downregulated), 674 (T1-NP vs. T2-NP: 286 upregulated and 388 downregulated), 341 (T1-P vs. T1-NP: 172 upregulated and 169 downregulated), and 379 (T2-P vs. T2-NP: 172 upregulated and 207 downregulated) DEGs in the comparison groups (Appendix A). Functional identification and analysis of these genes showed that they were involved in fruit pigment accumulation, texture changes, hormone synthesis, flavor and aromatic compound production.

#### 3.5.1. Expression Pattern Analysis of Color Synthesis-Related Genes

Previous studies have shown that the color depth of eggplant fruit is related to the content of anthocyanins [49], so we analyzed the DEGs associated with the essential steps of the anthocyanin synthesis pathway. In the photosensitive eggplant, a total of 35 genes (4 upregulated and 31 downregulated) related to eggplant fruit color were identified in the CK1-P vs. T1-P comparison group. From the transcriptome data, we found that the shading treatment significantly decreased the expression of *chalcone synthase* (*CHS*) (Figure 4B), *chalcone isomerase* (*CHI*) (Figure 4C), *dihydroflavonol 4-reductase* (*DFR*) (Figure 4D), and other enzymes related to anthocyanin synthesis, among which *anthocyanin synthase* (*ANS*) (Figure 4E), *anthocyanin-related transcription factor TT8* (Figure 4G)*,* and *3-O-glucosyltransferase* (*3GT*) (Figure 4H) were the most significantly downregulated. A total of 38 genes (33 upregulated and 5 downregulated) related to eggplant fruit color were identified in the T1-P vs. T2-P comparison group. The expression of genes related to anthocyanin synthesis tended to be significantly upregulated; these genes mainly included *chalcone synthase 2* (*CHS2*), *anthocyanin-related transcription factor TT8*, *anthocyanin synthase* (*ANS*), and *dihydroflavonol 4-reductase* (*DFR*). In addition, 31 DEGs were shared by the two groups (Figure 4A). Interestingly, the expression of these genes was downregulated after shading and upregulated after reillumination (Appendix A).

In the nonphotosensitive eggplant, a total of 25 (1 upregulated and 24 downregulated) DEGs related to eggplant fruit color were identified in the CK1-NP vs. T1-NP comparison group. Unlike in the CK1-P vs. T1-P comparison group, one *4-coumarate CoA ligase 2* (*4CL2*) (Figure 4F) gene in the CK1-NP vs. T1-NP comparison group showed an upregulated trend. Nevertheless, the other genes, such as *COP1* (Figure 4I), *F3′H* (Figure 4J), and *CHI3*, were all downregulated after the shading treatment. A total of 30 genes (22 upregulated and 8 downregulated) related to eggplant fruit color were identified in the T1-NP vs. T2-NP comparison group. We found that *CHS* and *4CL* were not significantly expressed, while the expression levels of *GT*, *TT8*, and *PAL* increased after reillumination. In addition, 12 genes related to eggplant fruit color were shared between the two comparison groups (Figure 4A) (Appendix A).

After 20 days of shading, 13 DEGs (12 upregulated and 1 downregulated) were detected in the T1-P vs. T1-NP comparison group, and only the *anthocyanin 5-O-glucosyltransferase* expression was downregulated, while *CHS2*, *transcription factor TT8* and *flavonoid 3’,5’-methyltransferase* (*FAOMT*) were most significantly upregulated. After reillumination, we identified only four DEGs (2 upregulated and 2 downregulated) related to eggplant color in T2-P vs. T2-NP. The expression of *anthocyanidin 3-O-Glucosyltransferase 5* (*GT5*) and *transcription factor MYB7* was downregulated, while the expression of two *transcription factors MYB15* was upregulated (Appendix A).

We found that the expression of 18 DEGs related to eggplant color was significantly downregulated after shading treatment in both eggplant types (Figure 4A), but these genes were downregulated more significantly in the photosensitive eggplant. After applying the shading treatment to the photosensitive eggplant, we found that *flavonoid 3,5-methyltransferase* (*F3’5’H*) (Figure 4K) and *ANS* decreased most significantly. After bagging treatment of the nonphotosensitive eggplant, we found that anthocyanin synthesis-related genes, such as *E3 ubiquitin-protein ligase COP1*, were downregulated most significantly, whereas *4-coumarate CoA ligase 2* (*4CL2*) was upregulated. In addition, we found that the essential genes involved in anthocyanin synthesis, such as *CHS*, *CHI*, *ANS*, *F3’H*, *DFR*, *F3’5’H*, and *MYB1* (Figure 4L), were almost not expressed in photosensitive eggplant peel after shading for 20 days but had a small amount of expression in nonphotosensitive eggplant peel, which resulted in the difference in appearance between the two kinds of eggplant. The results showed that there were great differences in the light response and pigment synthesis mechanism between the photosensitive eggplant and nonphotosensitive eggplant. The differential expression of these genes resulted in phenotypic differences between the photosensitive and nonphotosensitive eggplant fruits.

#### 3.5.2. Expression Pattern Analysis of Genes Related to Fruit Texture

The physical basis of fruit hardness stems from the mechanical strength of the material composing the cell wall [50]. Enzymes related to cell wall metabolism mainly include *β-galactosidase* (*β-Gal*), *endoglucanase* (*EG*), *pectinesterase* (*PE*), *polygalacturonase* (*PG*), and *pectin methylesterase* (*PME*), which play an important role in cell wall degradation [51,52,53]. Transcriptome analysis identified 14 genes (3 upregulated and 11 downregulated) related to eggplant fruit texture in the CK1-P vs. T1-P comparison group. It was evident that the texture-related genes in the photosensitive eggplant were especially downregulated after darkness; these genes mainly included *LOB domain-containing protein 12* (*LBD12*), *beta-amylase* 1 (*BMY1*), and *polygalacturonase 1* (*PG1*). A total of 35 genes (20 upregulated and 15 downregulated) related to eggplant fruit texture were identified in the T1-P vs. T2-P comparison group. Among these genes, *pectin acetylesterase* (*PAE*), *xyloglucan endotransglucosylase/hydrolase* (*XTH*), and *PG1* were most significantly expressed (Appendix A).

In the nonphotosensitive eggplant, the CK1-NP vs. T1-NP comparison revealed 27 DEGs (3 upregulated and 24 downregulated) related to eggplant fruit peel texture, among which *endochitinase* (*EC*), *LBD12*, *proline-rich protein 4* (*PRP4*), and 21 other genes were the most significantly downregulated. A total of 45 genes (27 upregulated and 18 downregulated) related to eggplant fruit peel texture were identified in the T1-NP vs. T2-NP comparison group, and *BMY1*, *pectinesterase* (*PME*), *PRP4*, and others were significantly differentially expressed (Appendix A).

After shading for 20 days, 13 DEGs (8 upregulated and 5 downregulated) related to eggplant texture were identified in the T1-P vs. T1-NP comparison group; among them, *PG1* and *PME12* were downregulated most significantly, while *EG18*, *beta-glucosidase* (*BGL*) and *EG1* were most significantly upregulated. After reillumination, we identified 10 DEGs (5 upregulated and 5 downregulated) related to eggplant texture in the T2-P vs. T2-NP comparison group. The expression of *WAT1-related protein* and *EC* was most significantly downregulated, while the expression of *PME* and *EC* was most significantly upregulated (Appendix A).

Transcriptome data showed that there were 10 DEGs in common after shading (Figure 5A), and these genes were significantly downregulated, except for *beta-amyrin synthase* (*BAS*), which was upregulated. In addition, 18 texture-related genes unique to each kind of eggplant were identified (Figure 5A). Among them, *endoglucanase 12* (*EG12*), *caffeic acid 3-O-methyltransferase 1* (*COMT1*), *beta-galactosidase 16* (*BG16*), and other genes were highly expressed in the photosensitive eggplant but not in the nonphotosensitive eggplant. In addition, *laccase-12* (*LAC12*), *cellulose synthase (CS)*, *LOB domain-containing protein 20* (*LBD20*), and other genes were differentially expressed in the nonphotosensitive eggplant but were not significantly different in the photosensitive eggplant. In general, most texture-related DEGs in eggplant were downregulated after shading and upregulated after reillumination. Interestingly, in contrast to the color results, most texture-related DEGs in the nonphotosensitive eggplant were downregulated more significantly after shading than those in the photosensitive eggplant. Figure 5B–I shows the variation in some major texture-related genes in the six comparison groups.

#### 3.5.3. Expression Pattern Analysis of Genes Related to Hormone Synthesis

Previous studies have shown that hormones such as auxin, abscisic acid (ABA), gibberellins, cytokinins, and ethylene can promote fruit growth and senescence [54]. Genes related to plant hormone synthesis and signal transduction mainly include *1-aminocyclopropane-1-carboxylate oxidase* (*ACO*) [55], *short chain reductase* (*SDR*) [56], *auxin-responsive protein SAUR* (Kant et al., 2009), and *zeatin O-xylosyltransferase* (*ZOX*) [57]. By comparing the transcriptome data from each group, we identified 22 (4 upregulated and 18 downregulated) genes related to hormones in the CK1-P vs. T1-P comparison group. These genes mainly include genes such as *ethylene-responsive transcription factor* (*ERF*)*, 1-aminocyclopropane-1-carboxylate oxidase homolog 1* (*ACO1*), *L-tryptophan--pyruvate aminotransferase 1* (*TAA1*), and *cytochrome P450 83B1* (*CYP83B1*). Moreover, a total of 36 genes (21 upregulated and 15 downregulated) related to hormones in eggplant fruit peel were identified in the T1-P vs. T2-P comparison group, among which *ethylene-responsive transcription factor 1B* (*ERF1B*), *ACO1*, *gibberellin-regulated protein 6* (*GASA6*), *zeatin O-xylosyltransferase* (*ZOX*), and other genes were most differentially expressed (Appendix A).

In the nonphotosensitive eggplant, we identified 31 (5 upregulated and 26 downregulated) genes related to hormones in the CK1-NP vs. T1-NP comparison group. Similar to the photosensitive eggplant CK1-P vs. T1-P comparison group, most of the hormone-related genes in the CK1-NP vs. T1-NP comparison group were also downregulated. Among them, *ERF*, *auxin-responsive protein SAUR65*, *methyl jasmonate esterase* (*MJE*), and other genes were the most significantly differentially expressed. A total of 43 genes (21 upregulated and 22 downregulated) related to hormones in eggplant fruit peel were identified in the T1-NP vs. T2-NP comparison group. Among them, *auxin-responsive protein IAA4*, *growth-regulating factor 1* (*GRF1*), *ERF*, *short-chain dehydrogenase/reductase* (*SDR*), and other genes were obviously differentially expressed (Appendix A).

After shading for 20 days, 8 DEGs (2 upregulated and 6 downregulated) related to eggplant hormones were identified in the T1-P vs. T1-NP comparison group; among them, *auxin-binding protein ABP19A* and protein NRT1 FAMILY were most significantly downregulated, while *gibberellin 2-beta-dioxygenase 1* (*GA2OX1*) and *CYP83B1* were most significantly upregulated. After reillumination, we identified 13 DEGs (6 upregulated and 7 downregulated) related to eggplant hormones in the T2-P vs. T2-NP comparison group. The expression of the *auxin-binding protein ABP19A* and *ACO* was most significantly downregulated, while the expression of *ERF1* and *ACS4* was most significantly upregulated (Appendix A).

After the shading treatment was applied to the photosensitive eggplant and nonphotosensitive eggplant, we found 13 common genes related to hormone activity in fruits from the transcriptome data (Figure 6A), among which only *GRF1* was upregulated in both photosensitive eggplant and nonphotosensitive eggplant. In addition, we identified 27 genes that were unique to either the photosensitive eggplant or nonphotosensitive eggplant (Figure 6A). Among them, *cytochrome P450 83B1, auxin responsive protein SAUR68*, *tryptophan aminotransferase-related protein* (*TAR*) and 6 other genes were significantly expressed in the photosensitive eggplant. Nevertheless, they were not differentially expressed in the nonphotosensitive eggplant. Moreover, *ERF038*, *gibberellin-regulated protein 6* (*GASA6*), *zeatin O-glucosyltransferase* (*ZOG1*), and *cytokinin dehydrogenase 1* (*CKX1*) were significantly expressed in nonphotosensitive eggplant. However, they were not differentially expressed in the photosensitive eggplant. The results showed that shading decreased the expression of most hormone-related genes in eggplant peel, while a few genes, such as *GASA* and *GRF*, were upregulated after shading. Interestingly, we also found that *ACO1*, a gene related to ethylene metabolism, was upregulated in photosensitive eggplant and downregulated in nonphotosensitive eggplant. Figure 6 shows the variation in some major hormone-related DEGs in the six comparison groups.

#### 3.5.4. Expression Pattern Analysis of Genes Related to Synthesizing Flavor and Aromatic Compounds

Fleshy fruits usually exhibit an accumulation of sugars, acids, and volatile compounds during ripening, resulting in the distinctive flavor and aroma of the fruit [58,59]. Common genes affecting flavor and aroma are *tropinone reductase* (*TR*) (Li et al., 2021), *alcohol dehydrogenase* (*ADH*) [60], *cinnamyl alcohol dehydrogenase* (*CAD*) [61], and (+)-*neomenthol dehydrogenase* ((+)-*ND*) [62]. In the CK1-P vs. T1-P comparison group for photosensitive eggplant, we identified 27 (10 upregulated and 17 downregulated) DEGs related to flavor and aromatic compounds in eggplant fruit peel. Among these genes, shading treatment effectively inhibited *pollen-specific leucine-rich repeat extensin-like* (*PEX*), *beta-amyrin 28-oxidase* (*CYP716A15*), *tropinone reductase* (*TR*), and *8-hydroxygeraniol dehydrogenase* (8HGO) expression. A total of 40 genes (31 upregulated and 9 downregulated) related to flavor and aroma were identified in the T1-P vs. T2-P comparison group; these genes mainly included (+)-*neomenthol dehydrogenase* ((+)-*ND*), *aldehyde dehydrogenase* (*ALDH*), *cinnamyl alcohol dehydrogenase 1* (*CAD1*) and *phenylalanine ammonia-lyase* (*PAL*) (Appendix A).

In the CK1-NP vs. T1-NP comparison group for nonphotosensitive eggplant, we identified 39 genes (6 upregulated and 33 downregulated) related to eggplant fruit peel flavor and aroma. These genes mainly included *CAD1*, *PEX4*, *TR*, and *ALDH*. Similarly, a total of 45 genes (32 upregulated and 13 downregulated) related to flavor and aroma were identified in the T1-NP vs. T2-NP comparison group. These genes mainly included *SWEET*, *TR,* and *ALDH* (Appendix A).

After shading for 20 days, 4 DEGs (0 upregulated and 4 downregulated) related to eggplant flavor and aroma were identified in the T1-P vs. T1-NP comparison group, among which *sugar transport protein 8* and *bidirectional sugar transporter SWEET10* were downregulated most significantly. After reillumination, we found 6 DEGs (6 upregulated and 7 downregulated) related to eggplant flavor and aroma in the T2-P vs. T2-NP comparison group. The expression of *TR* and *sugar transport protein 8* was most significantly downregulated (Appendix A).

By analyzing the transcriptome data from the fruit peels of photosensitive and nonphotosensitive eggplant, we identified 17 common flavor- and aroma-related genes in these eggplant fruits after shading treatment (Figure 7A). Among these genes, *alpha-farnesene synthase* (*AFS*), *aldehyde dehydrogenase family 16 member* (*ALDH16*), and *nucleotide-diphospho-sugar transferase family protein* (*At4g15970*) were upregulated in the peel of photosensitive eggplant but downregulated in the peel of nonphotosensitive eggplant. We also identified 25 genes that were unique to each kind of eggplant. Among them, *SWEET11*, *TR, ALDH2B7,* and other genes were significantly expressed in photosensitive eggplant but not in nonphotosensitive eggplant. At the same time, *lipoxygenase 6* (*LOX6*), *SWEET1*, *CAD1*, and other genes were differentially expressed in nonphotosensitive eggplant but not in photosensitive eggplant. Moreover, most of the differentially expressed genes related to aroma and flavor were more highly expressed in photosensitive eggplant than in nonphotosensitive eggplant after shading. Figure 7B–I shows the variation in some major flavor- and aroma-related genes in the six comparison groups.

## 4. Discussion

In recent years, horticultural facilities in China have undergone rapid development, especially in northern China. The area of solar greenhouses is increasing annually. These greenhouses are mainly used for the off-season production of winter and spring vegetable species, one of which is eggplant. In northern China, eggplant is often exposed to low light during its growth because of the short days and long nights in the winter. Low irradiance is one of the key factors causing the difference in color, texture, hormone, flavor, and aroma of eggplant peel, among which the color change is the most obvious, which greatly affects the production and sales of eggplant. With the continuous increase in global eggplant production, this drawback has been highlighted. Therefore, in this experiment, photosensitive and nonphotosensitive eggplant materials were bagged to exclude light [43]. The peel color of eggplant is an important breeding target trait. The appearance color of eggplant peel is positively correlated with the content of anthocyanin in the peel. Therefore, it is of great significance to study the content, composition, and synthesis pathway of eggplant anthocyanin for the study of eggplant peel color. In the production of eggplant, photosensitive eggplant often showed poor coloring under the condition of insufficient illumination, while nonphotosensitive eggplant could have good coloring under the condition of insufficient illumination. The physiological characteristics of the two kinds of eggplant under low-light and multiple light conditions were studied. Finally, the results showed that the experimental treatment affected the accumulation of transcript of key enzymes related to fruit color, texture, flavor, and aroma and the expression of critical genes and transcription factors related to hormone pathways during the eggplant ripening process (Figure 8).

The study of purple eggplant fruit peel has been critical, and different types and color intensities of purple eggplant organs have attracted researchers’ attention; this coloration is mainly related to anthocyanins. Bagging can effectively improve fruit quality and is a suitable method for studying anthocyanin biosynthesis and related gene expression [63,64]. Currently, bagging is an important part of fruit and vegetable cultivation, including that of apples, pears, peaches, grapes, and loquats, in many countries, such as China, Japan, Australia, and the United States [65]. Through bagging, we found that the color of the photosensitive eggplant fruits became white, while the color of the nonphotosensitive eggplant fruits slightly faded. Cluster analysis showed that most of the genes involved in anthocyanin biosynthesis and the flavonoid biosynthesis pathway were downregulated in the bagged eggplant fruits. Nevertheless, the downregulation was more evident in the photosensitive eggplant (Figure 3). This difference in gene regulation suggests that the mechanisms of anthocyanin synthesis in photosensitive eggplant and nonphotosensitive eggplant vary under different light conditions. Transcriptome comparison data showed that the genes related to anthocyanin synthesis, such as *COP1*, *PAL*, *AN3*, *F3′H*, *RT*, *ANT*, *DFR*, *CHS*, *CHI*, and *MYB1*, were downregulated in the two kinds of eggplant, and the gene expression changes were more significant in the photosensitive eggplant. Many studies have shown that upregulation or downregulation of these genes can lead to changes in anthocyanin accumulation. After bagging litchi, researchers found that *COP1* enhances ubiquitination activity by targeting *HY5* and other transcriptional activators, thereby negatively regulating anthocyanin accumulation [66,67]. Overexpression of the *CHS* gene in tobacco results in a change in the anthocyanin content [68], and inhibition of *DFR* expression in sweet potato can reduce the anthocyanin content in the roots and stems [69]. Changes in *PAL* expression can lead to anthocyanin content changes in the spear-like parts of the apex and base of white asparagus after harvest [70]. Overexpression of *CHS*, *CHI*, and *DFR* in Arabidopsis increases the pigment content in the stems and leaves [30]. In addition, in *Arabidopsis thaliana*, *AtAN3* negatively regulates *AtCOP1* expression at the transcriptional level, affecting anthocyanin biosynthesis [71]. Overall, the accumulation of anthocyanins in purple eggplant fruit peel was consistent with previous findings. The different colors of the two kinds of eggplant after shading treatment were related to the different expression levels of genes involved in anthocyanin synthesis.

The E3 ubiquitin-protein ligase *COP1* acts as a light-induced morphogenetic switch [72]. Under light conditions, *COP1* activity is inhibited by photoreceptors, and *COP1* is exported from the cytoplasm, inducing a response to light [40]. Datta et al. [73] found that *COP1* is localized in the nucleus in a dark environment and can induce photomorphogenesis transcription factor ubiquitination and degradation. In apple, *MdCOP1* can interact with *MdMYB1* to regulate anthocyanin biosynthesis [74]. Our study showed that under dark conditions, the photosensitive eggplant material could not perceive light, and the photoregulated switch *SmCOP1* was “off”; additionally, transcription factors such as *SmMYB1* were not expressed, anthocyanins could not be synthesized, and the color of the eggplant fruit peel became white. In the nonphotosensitive eggplant fruits, although the photoregulated switch *SmCOP1* was “off”, the expression of the downstream transcription factor *SmMYB1* was activated through other photoregulatory switches, which enabled the synthesis of anthocyanins and a change in eggplant color. In addition, the structural gene *SmANS* functions in converting colorless forms of anthocyanins into colored ones, which is crucial for the coloration of flowers, fruits, and leaves. Previous studies have shown that inhibition of the expression of *SmANS* in calla lily (*Zantedeschia aethiopica* L.) and butterfly grass (*Torenia* L.) can produce colorless anthocyanins, resulting in white plants [75]. Our research results are similar. Under shading treatment, *ANS* was not expressed in the photosensitive eggplant, while it was expressed in the nonphotosensitive eggplant, which may be the main reason why the fruit peel of photosensitive eggplant was white after shading treatment while the peel of nonphotosensitive eggplant was mauve.

The change in fruit texture is also an important evaluation criterion in the process of fleshy fruit ripening and senescence [76]. Many studies have shown that enzymes such as *beta-galactosidase* (*β-Gal*), *pectin acetylesterase* (*PAE*), *cellulase* (*CL*), and *polygalacturonase* (*PG*) play essential roles in plant cell wall synthesis and degradation [50,77]. The expression levels of *LBD12*, *PG*, *beta-amylase 3* (*BAM3*), and *cinnamoyl-CoA reductase* (*CCR*) significantly decreased after shading treatment, and the degree of downregulation of texture-related genes in photosensitive eggplant was more significant. The expression levels of the *beta-galactosidase 16* (*β-Gal16*), *endoglucanase 3* (*EG3*)*, endoglucanase 8* (*EG8*), and *sucrose synthase 5* (*SUS5*) genes were downregulated after reillumination, which was consistent with the results of previous studies. Bu et al. [78] found that UV-C treatment inhibited the activities of *cellulase* (*Cel*), *polygalacturonase* (*PG*), and *expansin* (*Exp*) enzymes. Similarly, Barka et al. [79] found that UV-C treatment reduced the activity of tomato cell wall degradation-related enzymes, thereby delaying the softening of tomato fruits. In conclusion, we found that shading treatment not only altered the color of the fruit peels of photosensitive eggplant and nonphotosensitive eggplant but also affected the texture of eggplant fruits by influencing the activities of cell wall degradation-related enzymes. In contrast, the expression changes of texture-related genes in photosensitive eggplant were more significant after shading treatment.

A variety of flavor compounds related to terpenoids, sesquiterpenoids, triterpenoids, sugars, phenylpropionic acids, and unsaturated fatty acids have been identified in eggplant. Hanifah et al. [80] found that terpenoids were the most abundant in eggplant according to GC–MS analysis. Our study found that the *PEX4*, *phosphoenolpyruvate translocator 2* (*PPT2*), *8HGO*, and *TR2* genes related to plant flavor and aroma were significantly inhibited in eggplant fruit peel after shading treatment. The expression of the *PAL*, *TR*, *SDR1*, and *glyceraldehyde-3-phosphate dehydrogenase* (*GAPC*) genes was upregulated after reillumination. In addition, there were some genes that were downregulated after shading treatment and upregulated after reillumination. For example, *BAS1* was upregulated in nonphotosensitive eggplant after shading treatment, and *glutamine synthetase-like isoform X1* (*GLN1-1*), (+)-*neomenthol dehydrogenase* ((+)-*ND*), and other genes were downregulated after reillumination in photosensitive eggplant. In conclusion, these results suggest that light may affect the synthesis and breakdown of eggplant fruit peel metabolites by regulating the expression of flavor- and aroma-related enzymes, thus affecting changes in fruit flavor and aroma.

The effects of hormones on fruit development persist throughout the whole process, from flowering to fruit ripening. The regulatory impact of hormones on plants is complex and comprehensive [81]. The light signaling pathway affects hormone synthesis and signal transduction through interactions with various hormone pathways under dense planting and low-light environments. Different levels of ethylene can regulate the growth and development of eggplant fruits [82,83]. *ACC* is a precursor of ethylene and promotes plant senescence [84]. Jiang et al. [19] found that the content of *ACC* in tomato leaves increased under low-light treatment, indicating that low-light treatment may accelerate the senescence of tomato leaves. The ethylene-related expression gene *ACO*, an *ACC* oxidase, is the last enzyme in the ethylene biosynthesis pathway. Our study found that *ACO* gene expression in eggplant fruits was downregulated after shading treatment and upregulated after reillumination. These results indicate that low levels of light are not conducive to the expression of ethylene-related genes. Zeatin is an active cytokinin that can promote cell division and regulate plant growth [85]. Jiang et al. [19] found that the decrease in stem diameter caused by low-light treatment may be due to a reduction in the endogenous zeatin concentration. In the present study, the expression levels of zeatin-related enzymes were downregulated in the photosensitive and nonphotosensitive eggplant fruit peel after shading treatment but upregulated after 5 days of reillumination. These results suggest that zeatin may have a regulatory effect in eggplant fruits under dark conditions. Darkness affects the complex interactions of hormones during the fruit ripening and senescence of different types of eggplant.

## 5. Conclusions

In this study, a comparative transcriptome analysis was used to study the molecular mechanism of dark-induced changes in the regulatory activity within fruit peels of photosensitive and nonphotosensitive eggplant. The results showed that shading treatment had many effects on the fruit peel of eggplant. Nevertheless, compared with the adverse impact on photosensitive eggplant, that on nonphotosensitive eggplant was weaker and was mainly reflected by the color, texture, and hormones in the fruit. By comparing the significant DEGs in each group, we identified the genes that may cause the differences between photosensitive eggplant and nonphotosensitive eggplant under shading treatment, which provided clues for understanding the molecular mechanism underlying the light–dark regulatory effects that differ between photosensitive and nonphotosensitive eggplant materials.

## Figures and Tables

**Figure 1 plants-11-02095-f001:**
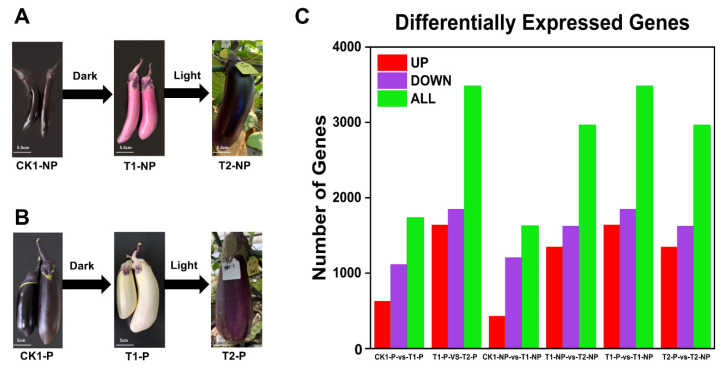
Changes in the appearance and genes of two eggplant varieties after bagging and illumination. (**A**) Changes in the appearance of nonphotosensitive eggplant after shading and illumination; (**B**) changes in the appearance of photosensitive eggplant after shading and illumination; and (**C**) changes in differentially expressed genes in the six comparison groups (both upregulated and downregulated).

**Figure 2 plants-11-02095-f002:**
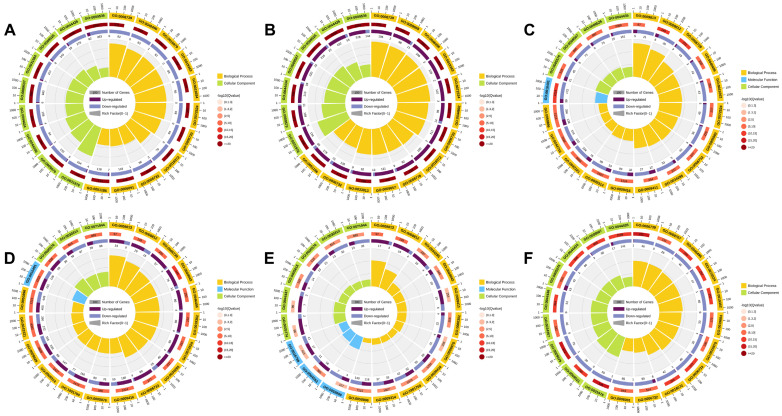
GO enrichment circle diagram of the six comparison groups. The first circle: enrichment of the first 20 GO terms; the coordinate ruler for the gene number is outside the circle. Different colors represent different ontologies. The second circle: the number and Q value of the GO term in the background gene. The more genes present, the longer the bar, the smaller the Q value, and the redder the color. The third circle: bar chart of the proportion of upregulated genes; dark purple represents the proportion of upregulated genes, and light purple represents the proportion of downregulated genes. Specific values are shown below. The fourth circle: Rich Factor value of each GO term (the number of different genes in the GO term divided by all the numbers) and background grid lines, where each grid represents 0.1 ((**A**) comparison group CK1-P vs. T1-P; (**B**) comparison group T1-P vs. T2-P; (**C**) comparison group CK1-NP vs. T1-NP; (**D**) comparison group T1-NP vs. T2-NP; (**E**) comparison group T1-P vs. T1-NP; and (**F**) comparison group T2-P vs. T2-NP).

**Figure 3 plants-11-02095-f003:**
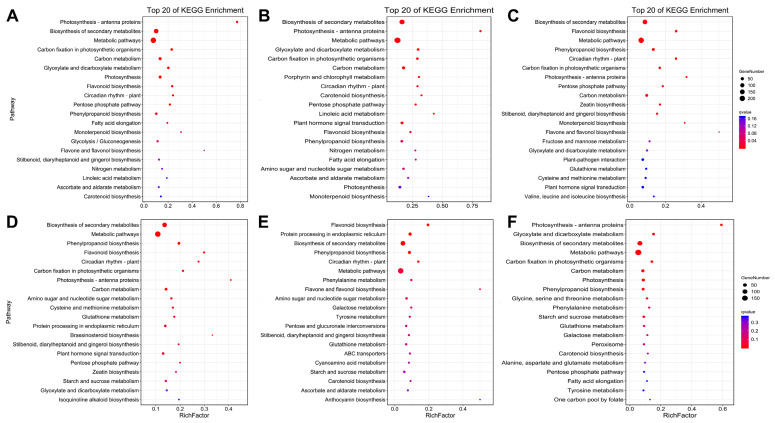
KEGG enrichment bubble plot. The first 20 pathways with the smallest Q values were used for the plot, with pathway as the ordinate and enrichment factor as the abscissa (the differentially expressed genes in this pathway are divided into groups based on the numbers shown). The size indicates the number, and the redder the color is, the smaller the Q value ((**A**) comparison group CK1-P vs. T1-P; (**B**) comparison group T1-P vs. T2-P; (**C**) comparison group CK1-NP vs. T1-NP; (**D**) comparison group T1-NP vs. T2-NP; (**E**) comparison group T1-P vs. T1-NP; and (**F**) comparison group T2-P vs. T2-NP).

**Figure 4 plants-11-02095-f004:**
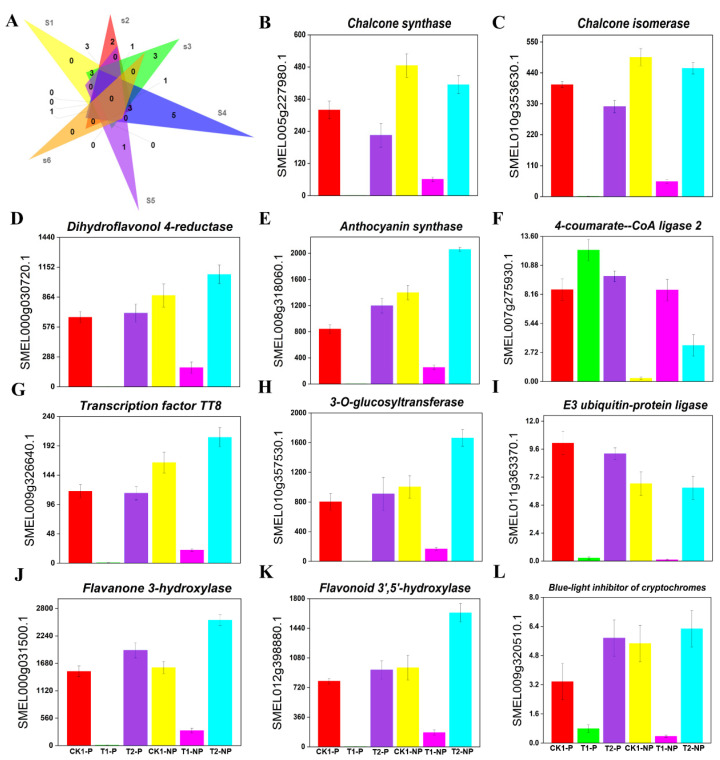
Effects of shading and light on gene expression and signal transduction related to eggplant peel color. (**A**) Venn diagram showing differentially expressed genes related to eggplant peel color in the six comparison groups (S1: CK1-P vs. T1-P; S2: T1-P vs. T2-P; S3: CK1-NP vs. T1-NP; S4: T1-NP vs. T2-NP; S5: T1-P vs. T1-NP; S6: T2-P vs. T2-NP); (**B**) *Chalcone isomerase* (*CHS*); (**C**) *Chalcone isomerase* (*CHI*); (**D**) *Dihydroflavonol 4-reductase* (*DFR*); (**E**) *Anthocyanin synthase* (*ANS*); (**F**) *4-Coumarate--CoA ligase 2* (*4CL2*); (**G**) *Anthocyanin-related transcription factor TT8*; (**H**) *3-O-glucosyltransferase* (*3GT*); (**I**) *E3 ubiquitin-protein ligase* (*COP1*); (**J**) *Flavanone 3-hydroxylase* (*F3′H*); (**K**) *Flavonoid 3′,5′-hydroxylase* (*F3′5′H*); and (**L**) *Transcription factor MYB1*. ((**B**–**L**): The *x*-axis represents different eggplant groups, and the *y*-axis represents FPKM values); the error bar represents the degree of dispersion of the gene expression data itself, as follows.

**Figure 5 plants-11-02095-f005:**
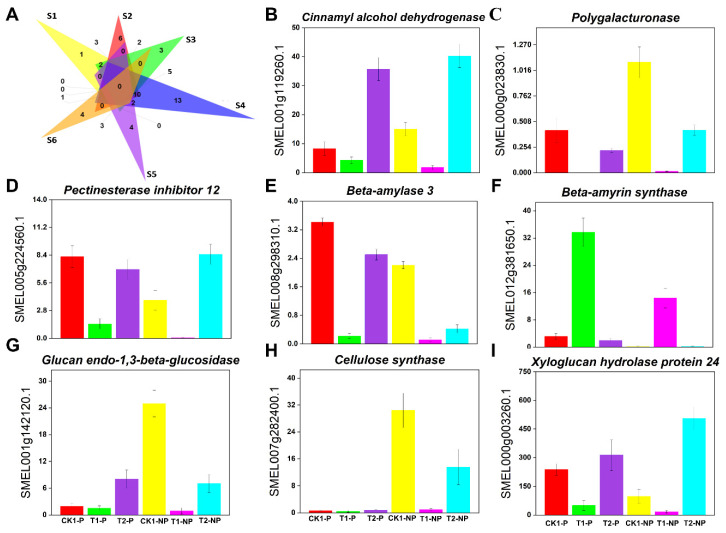
Effects of shading and light on gene expression and signal transduction related to eggplant peel texture. (**A**) Venn diagram showing differentially expressed genes related to eggplant peel texture in the six comparison groups (S1: CK1-P vs. T1-P; S2: T1-P vs. T2-P; S3: CK1-NP vs. T1-NP; S4: T1-NP vs. T2-NP; S5: T1-P vs. T1-NP; and S6: T2-P vs. T2-NP); (**B**) *Cinnamyl alcohol dehydrogenase* (*CAD*); (**C**) *Polygalacturonase* (*PG*); (**D**) *Pectinesterase inhibitor 12* (*PME12*); (**E**) *Beta-amylase 3* (*BAM3*); (**F**) *Beta-amyrin synthase* (*BAS*); (**G**) *Glucan endo-1,3-beta-glucosidase* (*GLUB*); (**H**) *Cellulose synthase* (*CES*); and (**I**) *Xyloglucan hydrolase protein 24* (*XTH24*). ((**B**–**I**): The X-axis represents different eggplant groups, and the Y-axis represents FPKM values).

**Figure 6 plants-11-02095-f006:**
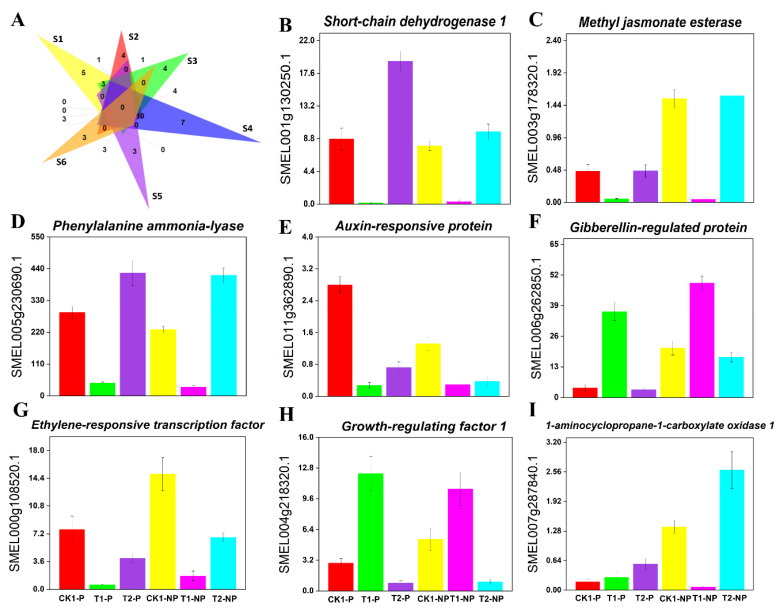
Effects of shading and light on gene expression and signal transduction related to eggplant peel hormones. (**A**) Venn diagram showing differentially expressed genes related to eggplant peel hormones in the six comparison groups (S1: CK1-P vs. T1-P; S2: T1-P vs. T2-P; S3: CK1-NP vs. T1-NP; S4: T1-NP vs. T2-NP; S5: T1-P vs. T1-NP; S6: T2-P vs. T2-NP); (**B**) *Short-chain dehydrogenase 1* (*SDR1*); (**C**) *Methyl jasmonate esterase* (*MJE*); (**D**) *Phenylalanine ammonia-lyase* (*PAL*); (**E**) *Auxin-responsive protein*; (**F**) *Gibberellin-regulated protein* (*GASA*); (**G**) *Ethylene-responsive transcription factor* (*ERF*); (**H**) *Growth-regulating factor 1* (*GRF1*); and (**I**) *1-aminocyclopropane-1-carboxylate oxidase* (*ACO*). ((**B**–**I**): The X-axis represents different eggplant groups, and the Y-axis represents FPKM values).

**Figure 7 plants-11-02095-f007:**
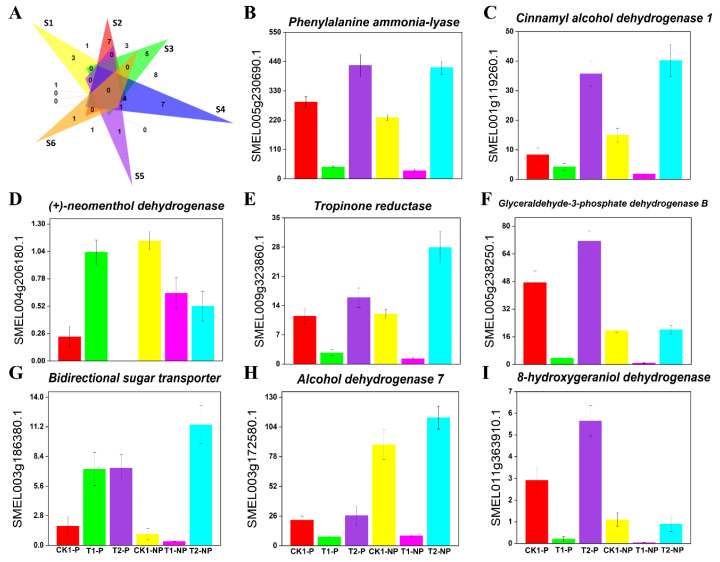
Effects of shading and light on gene expression and signal transduction related to eggplant peel flavor and aromatic compounds. (**A**) Venn diagram showing differentially expressed genes related to eggplant peel flavor and aroma in the six comparison groups (S1: CK1-P vs. T1-P; S2: T1-P vs. T2-P; S3: CK1-NP vs. T1-NP; S4: T1-NP vs. T2-NP; S5: T1-P vs. T1-NP; S6: T2-P vs. T2-NP); (**B**) *Phenylalanine ammonia-lyase* (*PAL*); (**C**) *Cinnamyl alcohol dehydrogenase 1* (*CAD1*); (**D**) (+)-*neomenthol dehydrogenase* ((+)-*ND*); (**E**) *Tropinone reductase (TR)*; (**F**) *Glyceraldehyde-3-phosphate dehydrogenase B* (*GAPB*); (**G**) *Bidirectional sugar transporter* (*SWEET*); (**H**) *Alcohol dehydrogenase 7* (*AD7*); and (**I**) *8-Hydroxygeraniol dehydrogenase* (*8HGO*). ((**B**–**I**): The X-axis represents different eggplant groups, and the Y-axis represents FPKM values).

**Figure 8 plants-11-02095-f008:**
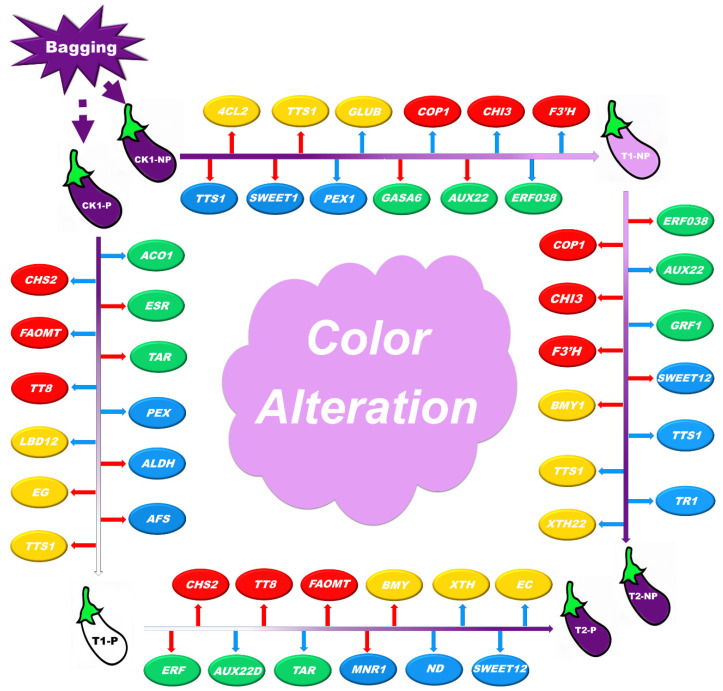
Model of genetic changes in response to shading and light during eggplant peel ripening. By comparing the differentially expressed genes between the two varieties of eggplant under shading and illumination, we found differences in several key genes related to color, texture, plant hormones, flavor, and aroma during eggplant peel ripening. (The blue arrow represents downregulation; the red arrow represents upregulation; the genes in the red icon are associated with eggplant color; the genes in the yellow icon are associated with eggplant texture; the genes in the green icon are associated with eggplant plant hormones; the genes in the blue icon are associated with eggplant flavor and aroma).

**Table 1 plants-11-02095-t001:** Anthocyanin content of photosensitive and nonphotosensitive eggplant in control group.

Group Name	Photosensitive Eggplant Peel	Nonphotosensitive Eggplant Peel
CK1-1	20.0 mg/kg FW	67.3 mg/kg FW
CK1-2	22.1 mg/kg FW	47.4 mg/kg FW
CK1-3	19.1 mg/kg FW	79.1 mg/kg FW

## Data Availability

All data included in the main text.

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
