# Peer review of "Determining the Effects of Light on the Fruit Peel Quality of Photosensitive and Nonphotosensitive Eggplant"

_plants, 2022, doi:10.3390/plants11162095_

Round 1
Reviewer 1 Report
Sang et al. studied the expression of genes in shaded and non-shaded photosensitive and nonphotosensitive eggplant fruits. While the research is within the scope of Plants, I cannot suggest the acceptance of the manuscript mainly because of the absence of metabolite-related parameters and of the discussion part which is mostly descriptive failing to focus on the physiological phenomenon. Moreover, the absence of illumination does not equal to the low light conditions (stated by the authors throughout the manuscript.
Lines 16, 17 revise, the phrases are awkward
Lines 46-56 The statement is based on a wrong basis that cellulases are important for fruit quality and storage, in plants other enzymes govern the physiological processes during late maturation and during the postharvest life. Moreover, the paragraph is unconnected without much coherence (this is true for all the introduction part).
Lines 83-84 and 88-89. A contradiction exists here.
Materials and methods.
The authors should at least try to estimate the concentrations of the critical members of the anthocyanin and anthocyanin-related metabolites. Moreover, the inclusion of texture-related data would have been greatly beneficial for this research.
Describe the abbreviations of the treatments in the Plant materials and treatment part (not at the fig. 1 legend)
Describe in higher detail the experimental design and the number of replications per treatment
Provide information about the material of the bags used for the treatments
Provide information about the depth of the peels used for this experiment
Discussion
No light does not equal to low light regarding the coloration of eggplants. Please revise the argument.
The discussion part is rather descriptive and the conclusions drown by the results while valid, they are expected and do not offer something remarkable to the international literature especially given the absence of metabolite and texture-related parameters.
Reviewer 2 Report
Very readable paper easy to understand because of your detailed explanation of your results. Very innovative approach to elucidate the role of light on eggplant physiology related to mature fruits.
Reviewer 3 Report
The authors set out to investigate the effect of light on the quality of eggplant grown in greenhouse conditions. They focused on developing features essential for good sales, i.e. the quality of the fruit itself and its peel colour. They chose a photosensitive line and a non-photosensitive line for their research.
However, it should be noted in the title that all results come only from the transcriptome analysis. It would also be helpful to note that this does not necessarily translate into the activity of individual biosynthetic pathways. Remember, however, that most enzymes are post-translational regulated.
Perhaps it would be helpful to quantify individual peel pigments under the conditions tested to see if these results match the level of the transcripts.
I miss a bit of a critical approach to the authors' results.
I like the work, especially Figure 8 is a good summary of the analysis of many results.
If I have reservations, it is only to the legibility of the figures. I understand that the article has an online version, but it also has a classic pdf and figures 2 and 3 cannot be read in it. However, Authors can enlarge them and arrange them in two columns instead of three.
Overall the work is valuable and well written as far as I can judge it. After minor corrections, mainly those pictures and notes about transcriptomics, the work is ready for publication.
Reviewer 4 Report
General comments:
The paper is very interesting and important dealing with the effects of eggplant fruit bagging on its quality. However some questions needs to be better explained for the readers and the key results should be confirmed by the most quantitative method.
Detailed comments:
1) Line 3 and line 15 - please check the corresponding author, it seems like the e-mail address belongs to the author not indicated as corresponding (Jinhua Zuo).
2) Line 24: darkness or low-light?
3) Line 44: "most plants avoid darkness" - better: "autotrophic plants avoid darkness"
4) Line 44/45: phototropism can be positive or negative, so this is a simplification, please reconsider.
5) In the introduction, starting from line 89 three types of eggplant accessions are mentioned but criteria for this classification are not sufficiently explained. What do photosensitive, nonphotosensitive and semiphotosensitive type mean?
6) Material and methods: The section 2.5 is not clear. What are replications, what does "each value is presented as the mean+-sd (why not se?)" means. Are there DEGs from different experiments? DEGs are differential values thus which way differences between treatments can be tested by the LSD test? The statistical treatments must be clearly explained and match the results presented in the manuscript.
7) The genes selected for figure 4 are proper illustration for the key genes shaping egg-plant peel color. However, these data are not sufficiently reliable without confirmation with quantitative method (qPCR). I think that these analysis should be performed. It is also not clear what are error bars in this figure (see the methodology of transcriptom analysis described before). The same refers to the figures 5-7.
8) Figure 8 is very helpful but should be explained in the caption. The relationships show in it doesn't look like networks and the change of colour from eg. purple to white (to the left) triggered expression of genes (This is what the figure shows). This must be changed.
Round 2
Reviewer 1 Report
After the inclusion of the anthocyanin levels and the revisions made by the authors, the article can be accepted.
Reviewer 4 Report
The manuscript was sufficiently improved and can be published in the current version.